# `AutoTrial`: Prompting Language Models for Clinical Trial Design

**Zifeng Wang**
UIUC
Urbana, IL, USA
zifengw2@illinois.edu

**Cao Xiao**
GE Healthcare
Seattle, WA, USA
cao.xiao@ge.com

**Jimeng Sun**
UIUC
Urbana, IL, USA
jimeng@illinois.edu

## Abstract

Clinical trials are critical for drug development. Constructing the appropriate eligibility criteria (i.e., the inclusion/exclusion criteria for patient recruitment) is essential for the trial's success. Proper design of clinical trial protocols should consider similar precedent trials and their eligibility criteria to ensure sufficient patient coverage. In this paper, we present a method named `AutoTrial` to aid the design of clinical eligibility criteria using language models. It allows (1) controllable generation under instructions via a hybrid of discrete and neural prompting, (2) scalable knowledge incorporation via in-context learning, and (3) explicit reasoning chains to provide rationales for understanding the outputs. Experiments on over 70K clinical trials verify that `AutoTrial` generates high-quality criteria texts that are fluent and coherent and with high accuracy in capturing the relevant clinical concepts to the target trial. It is noteworthy that our method, with a much smaller parameter size, gains around 60% winning rate against the GPT-3.5 baselines via human evaluations.

## 1 Introduction

Generative large language models (LLMs) are drawing attention due to their ability to create coherent and human-like text documents. Clinical trial design documents are written at the planning stage of the drug development process, which is crucial for the success of the trial. However, it can be challenging even for experienced professionals: around 57% trial protocols have at least one substantial amendment in eligibility criteria (CSDD, 2016). The suboptimal trial design may cause insufficient recruitment, severe adverse events, or insignificant efficacy, thus inducing huge financial losses and time waste. Each amendment will further cause millions of dollars in loss and months of delays.

In this paper, we propose to generate the eligibility criteria for clinical trials in natural language using LLMs, with the solution focusing on the following aspects.

- **Comprehending instructions**. The LLM will be prompted with key information about a trial, such as the target conditions and treatments, and the additional instruction to generate the criteria for participant recruitment. This process requires the employed LLM to comprehend the input and adapt to the input instruction to generate precise eligibility criteria that meet the specified objectives. It also necessitates the domain knowledge about clinical trials stored in LLMs.

- **Referring to prior studies**. A thorough literature search is important for human experts to design clinical trials (Chew, 2019). Similarly, the employed LLMs should be able to leverage the context information, such as retrieved eligibility criteria from prior successful studies, as the reference to generate better trial design.

- **Rationalizing the generation**. LLMs should offer the rationale behind the generated criteria, which is important for clinical experts to understand and adopt the generation results in practice.

In this paper, we propose to augment clinical trial design using LLMs, motivated by the evidence that LLMs can act as implicit knowledge bases (Petroni et al., 2019; Taylor et al., 2022). Our method is equipped with instruction tuning for trial protocol design and explicit supervision for producing grounding rationales. This is enabled with the following technical features:

- **Instruction prompting for adapting expert instructions**. Instruction tuning for granular control over the generated criteria to follow diverse user intentions.

- **Scalable and efficient knowledge expansion**. A combination of 1) external memory for a dense re-

triever and 2) internal memory for neural prompting, which is amenable to updating the model incrementally as new data are available.

- **Explicit supervision for generating grounding rationales**. Adaption of LLMs with a reasoning capability through a supervised paradigm, making the generated criteria more transparent and interpretable.

Our `AutoTrial` is the first model that utilizes LLMs for automating clinical trial design. It represents an important step toward using AI to facilitate clinical trial design. The rest of the paper is organized as follows: In §2, we review related work. In §3, we dive into the proposed method in detail. In §4, we present the experiment results. It is noteworthy that `AutoTrial` is proven to generate accurate criteria (with precision 0.91, recall 0.92, F1 0.91, and Jaccard score of 0.84 in clinical accuracy evaluation) while almost all baselines get less than 0.5 in these metrics. Moreover, our method reaches around 60% winning rate against GPT-3.5 [1] in trial design tasks via human evaluation. Finally, in §5, we conclude and discuss future work.

## 2 Related Work

### 2.1 Large Language Models

Large language models pre-trained on web-scale text data exhibit extraordinary emergent capability in a diverse set of natural language processing (NLP) tasks (Kaplan et al., 2020; Brown et al., 2020). It was recently witnessed that LLMs can be further tuned to align with human preferences through instruction tuning (Chung et al., 2022; Wang et al., 2022) and reinforcement learning from human feedback (RLHF) (Ouyang et al., 2022; Yuan et al., 2023; Dong et al., 2023).

Despite the remarkable capabilities of large language models (LLMs) trained on general text corpus, they often face challenges when generating highly domain-specific tasks unless they undergo additional tuning. Research has shown that even a "small" 300M LM, with instruction tuning, can outperform LLMs with over 100B parameters (Yasunaga et al., 2022). This finding encourages the efforts to develop customized expert LLMs by performing instruction tuning on domain-specific datasets, e.g., clinical notes and scientific publications (Singhal et al., 2022). In this work, we

are the first to develop LLMs focusing on trial design through a mixture of techniques, including instruction tuning, evidence-grounded generation, and supervised learning for rationale generation.

### 2.2 Clinical Trial Design

The clinical trial design is a new research topic for the NLP community, and there are only a few works related to clinical trial design, either focusing on trial feature embedding or trial design evaluation. For trial feature embedding, Marshall et al. (2017) extracted text pieces that describe the key trial characteristics as a summary report. More recently, Wang and Sun (2022) developed a self-supervised document model for dense retrieval for clinical trials. For trial design evaluation, Kim et al. (2021) manually adjusted criteria to broaden patient accrual and assess the influence of criteria, while Liu et al. (2021) utilized Shapley scores (Lundberg and Lee, 2017) to estimate the change of hazard ratio of the included oncology patients when removing each criterion. Despite these efforts, no existing work focuses on clinical trial design automation.

## 3 Method

`AutoTrial` utilizes a decoder-based architecture for generating a target criterion based on input trial synopsis and manual instructions. The training process consists of two stages: **pretraining** and **finetuning**.

- During the pretraining stage, the model is trained on a large corpus of trial documents in order to learn to reason through multiple steps and mimic the retrieved input criteria exemplars.

- In the finetuning stage, the model is trained to generate the target criterion according to the input instructions. For example, an instruction `<age>` that urges the model to populate the criterion describing the participant's age requirement.

It is noteworthy that the model can be extended to new instructions and trial exemplars without retraining. The flowchart is depicted in Fig. 1. We will elaborate on the details of the training and inference procedures of `AutoTrial` next.

### 3.1 Problem Setup

The generation model is represented by the function $f$, and generates a target criterion $\mathbf{y}_c$ based on input $\mathbf{x} = \{\mathbf{x}_s, \mathbf{x}_e, \mathbf{x}_r\}$. Here, $\mathbf{x}_s$ denotes trial

[1]Engine `gpt-3.5-turbo-0301`: https://platform.openai.com/docs/models/gpt-3-5

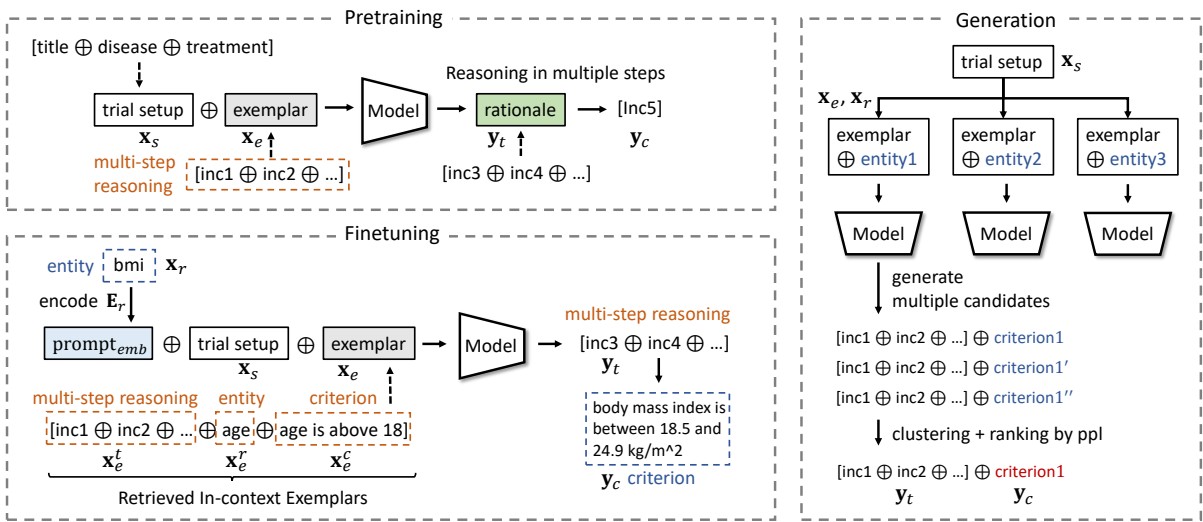

Figure 1: The workflow of the proposed `AutoTrial`. Step I: pre-train on unlabeled trial documents with prompts to mimic the multi-step reasoning. Step II: finetune the model to generate criteria under instructions. Step III: generate diverse target criteria by instructions with large-scale sampling plus clustering and ranking.

setups, which is a concatenation of the trial title, condition, and treatment, as the elements illustrated by Fig. 1. $\mathbf{x}_r$ denotes the discrete prompt describing the objective criterion, e.g., "bmi" prompts the model to generate the criterion for body mass index of the participants. $\mathbf{x}_e$ denotes the exemplars retrieved from relevant trials built for the in-context learning of LLMs. To this end, we formulate $\mathbf{x}_e = \{\mathbf{x}_e^t, \mathbf{x}_e^r, \mathbf{x}_e^c\}$, which contains the reasoning steps $\mathbf{x}_e^t$, e.g., a chain of criteria that leads to the target criteria, the targeting instruction $\mathbf{x}_e^r$, e.g., the objective to be set for recruiting patients, and the target criterion $\mathbf{x}_e^c$ that describes the requirement based on the instruction.

The model is also controlled by continuous prompt $\mathbf{h}_p$, which is specific to each type of instruction, e.g., the targeting entity that the criterion should contain. The model is trained to generate criteria $\mathbf{y}$ with multi-step reasoning: generating relevant criteria one by one and ultimately yielding the target criterion. Therefore, the generation process is expressed in Eq. (1),

$$\mathbf{y} = f(\mathbf{x}_s, \mathbf{x}_e, \mathbf{x}_r, \mathbf{h}_p). \tag{1}$$

Referring to the exemplar $\mathbf{x}_e$, the model outputs $\mathbf{y} = \mathbf{y}_t \oplus \mathbf{y}_c$, where $\mathbf{y}_t$ is the reasoning steps and $\mathbf{y}_c$ denotes the target criterion.

## 3.2 Hybrid Prompting

We opt to employ a hybrid of *discrete* and *neural* prompting to endow the model with the ability to generate criteria based on specific instructions.

### 3.2.1 Discrete Prompt

The discrete prompt is motivated by the prospect of in-context learning of LLMs (Wei et al., 2022), as the reasoning ability of LLMs can be enhanced via the input-output exemplars, e.g., the concatenation of a series of criteria $\mathbf{x}_e^t$, the target instruction $\mathbf{x}_e^r$, and the target criteria $\mathbf{x}_e^c$. We formulate the discrete prompts with specialized tokens:

1. **Trial Setup**: we wrap the introduction of trial setups, including title, disease, and treatment, using specialized tokens like `<title>`, `<disease>`, and `<treatment>`. The setup offers the basic context of a trial.

2. **In-context Exemplar**: we curate the exemplar that resembles the multi-step reasoning procedure: the model first generates a series of intermediate rationales that lead to the final answer. Concretely, the exemplar $\mathbf{x}_e = \{\mathbf{x}_e^t, \mathbf{x}_e^r, \mathbf{x}_e^c\}$ is retrieved from the *external knowledge store* and demonstrate as the template for the model outputs. $\mathbf{x}_e^t$ are many eligibility criteria wrapped by `<inc>` or `<exc>` indicating inclusion or exclusion criteria. $\mathbf{x}_e^r$ describes the instruction wrapped by `<statement>`, e.g., tell the model to generate a criterion describing the age requirement by "`<statement>` age". $\mathbf{x}_e^c$ is the target criterion wrapped by `<target>`, e.g., "`<target>` age is above 18 yrs old".

3. **Textual Instruction**: following the exemplar, $\mathbf{x}_r$ enforces the model to obey the instruction, wrapped by `<statement>` such as "`<statement>` gender".

The exemplars are stored in an external knowledge store providing an *open-book reference* that the model can refer to during the generation. It is built on the training data $\mathcal{C} = \{(\mathbf{x}_i, \mathbf{y}_i)\}_i^N$ that is amenable to edit, add or delete during the course of training and generating needless of retraining the model. We utilize a neural encoder $\mathcal{T}$ named Trial2Vec (Wang and Sun, 2022) that encodes trial setups $\mathbf{x}_s$ to dense embeddings, as $\mathbf{h}_s = \mathcal{T}(\mathbf{x}_s) \in \mathbb{R}^d$ that carry rich semantics of the trials. Consequently, the knowledge store is given by Eq. (2),

$$(\mathcal{K}, \mathcal{V}) = \{(\mathbf{h}_s, \mathbf{x}_e) \mid (\mathbf{x}, \mathbf{y}) \in \mathcal{C}\} \qquad (2)$$

with the embeddings serving as the keys and the exemplars as the values. Here, the vector-based search engine can be implemented for efficient exemplar retrieval on the fly.

### 3.2.2 Neural Prompt

Consider the embeded input tokens $\mathbf{x}_{<l} = \{x_1, \ldots, x_l\}$ as $\mathbf{H}_{<l} = \{\mathbf{h}_1, \ldots, \mathbf{h}_l\} \in \mathbb{R}^{l \times d}$, we prepend neural prompts to $\mathbf{H}_{<l}$ to get the prompted input $\widetilde{\mathbf{H}}_{<l} = \mathbf{h}_p \oplus \mathbf{H}_{<l}$. Formally, we create a set of instruction indices $\mathcal{I}$, the $i$-th instruction $\mathbf{x}_{r,i}$ is parameterized by Eq. (3),

$$\mathbf{h}_p = \text{MLP}(\mathbf{E}_r[i,:]), \qquad (3)$$

where $\mathbf{E}_r \in \mathbb{R}^{|\mathcal{I}| \times d'}$ is the trainable embedding matrix. $\mathbf{E}_r[i,:]$ indicates looking up the $i$-th row of the matrix; MLP : $\mathbb{R}^{d'} \mapsto \mathbb{R}^d$ projects the embedded instruction to the same dimension as $\mathbf{H}$.

The neural prompting is modular, meaning that it can be easily modified to incorporate additional instructions $\mathcal{I}'$ by simply extending the index set $\mathcal{I} = \{\mathcal{I}, \mathcal{I}'\}$ and the embedding matrix $\mathbf{E}_r = \{\mathbf{E}_r, \mathbf{E}'_r\}$ for those instructions. When the model is finetuned on new data, we can only update the instruction embedding $\mathbf{E}'_r$ while the rest of the model remains frozen. This allows the model to effectively learn to generate based on a broader range of instructions while minimizing the risk of catastrophic forgetting, i.e., the performance degradation on previous data.

### 3.3 Multi-stage Training

As described in §3.1, we have a dataset containing pairs of input instructions (denoted as $\mathbf{x}_r$) and corresponding criteria (denoted as $\mathbf{y}_c$). We extract clinical relations from the raw criteria to formulate the training and testing data, e.g., extracting

the relation "NYHA $\in$ {III, IV}" from the criteria "NYHA class is above II". However, the parser may not be able to extract all relevant instructions from all available trial documents. We hence propose to train our method in two stages: first pretraining on a large set of unlabeled trial documents and then finetuning on the processed dataset of instruction-criteria pairs. This approach allows us to make the most of the available data and facilitate the model performance.

**Pretraining**. We create a pretraining dataset

$$\mathcal{C}_{pre} = \{(\mathbf{x}_s, \mathbf{x}_e, \mathbf{y}_t, \mathbf{y}_c)_i\}_i^M, \qquad (4)$$

where the model $f$ is urged to generate $\mathbf{y} = \mathbf{y}_t \oplus \mathbf{y}_c$ in Eq. (1). The inputs comprise the trial setup $\mathbf{x}_s$ and the exemplar $\mathbf{x}_e$ which is also composed of multiple criteria. Drawing the inspiration from (Taylor et al., 2022), we decide to include prompts and special tokens in the pretraining stage. Specifically, we explicitly emphasize the step-by-step reasoning task by inserting the separate tokens <inc> and <exc> into $\mathbf{x}_e$ and $\mathbf{y}_t$, and the model is supervised to generate the intermediate rationales and yield the target criterion.

Our method is built based on decoder-based CLM (e.g., GPT2 (Radford et al., 2019)) where the decoder predicts $\mathbf{y}$ autoregressively. Denote the learned decoding distribution as $p_\theta(\cdot)$, the objective is the maximum log-likelihood estimation given by Eq. (5),

$$\mathcal{L}_{\text{MLE}} = -\frac{1}{L} \sum_{l=1}^{L} \log p_\theta(y_l | \mathbf{y}_{<l}, \mathbf{x}). \qquad (5)$$

where $\mathbf{y}_{<l}$ are tokens in $\mathbf{y}$ before the $l$-th token; $L$ is the total number of tokens in the target $\mathbf{y}$.

**Finetuning**. After pretraining, the model is finetuned on the dataset $\mathcal{C}$, and taught to follow the instruction when generating criteria. The inputs and outputs are described in Eq. (1). In addition to the MLE loss in Eq. (5), we apply a contrastive loss $\mathcal{L}_{\text{CL}}$ (Su et al., 2022) to enhance the model representation learning, as in Eq. (6),

$$\mathcal{L}_{\text{CL}} = \frac{1}{L \times (L-1)} \sum_{l=1}^{L} \sum_{j=1, j \neq l}^{L} \\ \max\{0, \rho - s(\mathbf{h}_{y_l}, \mathbf{h}_{y_l}) + s(\mathbf{h}_{y_l}, \mathbf{h}_{y_j})\}, \qquad (6)$$

where $\mathbf{h}_{y_l}$ is the embedding of token $y_l$, $\rho$ is the predefined margin, $s(\cdot)$ is the cosine similarity function. The finetuning loss combines the objectives

Table 1: The statistics of the used trial data.

|  | Train | Valid | Test |
| --- | --- | --- | --- |
| # trials | 54,703 | 6,079 | 15,195 |
| # inclusion | 153,169 | 17,145 | 42,269 |
| # exclusion | 128,310 | 14,581 | 35,247 |
| Avg inc length | 121.0 | 120.4 | 118.5 |
| Avg exc length | 148.3 | 153.0 | 145.2 |

from Eqs. (5) and (6) as given by Eq. (7),

$$\mathcal{L}_{\text{FT}} = \mathcal{L}_{\text{MLE}} + \mathcal{L}_{\text{CL}}. \qquad (7)$$

Note that the decoding distribution in the finetuning stage is $p_\theta(y|\mathbf{y}_{<l}, \mathbf{x}, \mathbf{h}_p)$ that differs from the one used in the pretraining shown in Eq. (5).

### 3.4 Generation

Denote the vocabulary by $\mathcal{V}$, we conduct top-k sampling repeatedly to acquire diverse candidates $\hat{\mathbf{Y}} = \{\hat{\mathbf{y}}_q\}_{q=1}^Q$, by Eq. (8),

$$y_l \sim p_\theta(y|\mathbf{y}_{<l}, \mathbf{x}, \mathbf{h}_p), \text{s.t. } y_l \in \mathcal{V}^{(k_s)}, \qquad (8)$$

where $\mathcal{V}^{(k_s)}$ is a subset of $\mathcal{V}$ that maximizes $\sum_{y \in \mathcal{V}^{(k_s)}} p_\theta(y|\mathbf{y}_{<l}, \mathbf{x}, \mathbf{h}_p)$, and $|\mathcal{V}^{(k_s)}| = k_s$. We further adopt clustering and ranking to select samples from the generated candidates. We first encode $\hat{\mathbf{y}}$ by Trial2Vec to dense embeddings $\mathbf{h}_{\hat{\mathbf{y}}}$ and apply k-means clustering with $k_q$ clusters. We then compute the perplexity (ppl) of each output $\hat{\mathbf{y}}_q$, and pick the sample with the minimum ppl in each cluster to form the final candidate set with $k_q$ samples. An example of the input and output of `AutoTrial` can be found in Table 5 in the appendix.

## 4 Experiment

We conduct extensive experiments to evaluate `AutoTrial` in the following aspects:

- The overall quality in terms of criteria-level and trial-level trial protocol generation.

- The capability of the continual update for new trials and instructions.

- The ablation analysis for the components of the proposed method.

### 4.1 Dataset

We collected clinical trial documents from ClinicalTrials.gov (NIH, 2023) and filtered out those without valid interventions, diseases, or titles, as well as those with void eligibility criteria. We extracted 75,977 clinical trials and applied Clinical Trial Parser (FAIR, 2022) to extract 751,258 medical relations from the eligibility criteria of these trials. The train-test split is shown in Table 1. For each trial, we sampled one criterion as the target and several others as input exemplars, resulting in 2,528,231 unique training samples out of 400K trials as the pretraining data. The validation and test trials were excluded from the pretraining data.

### 4.2 Evaluation Strategy

**Automatic Evaluation**. To evaluate the quality of the output criteria, which are expressed in natural language, we employ metrics from the NLG literature, including CIDEr (Vedantam et al., 2015), ROUGE-L (Lin, 2004), METEOR (Lavie and Agarwal, 2007), and BLEU (Papineni et al., 2002). These metrics allow us to assess the fluency and coherence of the generated criteria quantitatively.

We evaluate all the methods at the *criteria* level and *trial* level. At the *criteria* level, the model will generate each criterion separately, using the concatenated trial setup texts and the first three tokens of the targeting criteria as input. At the *trial* level, the model will take the concatenated trial setup texts as input and generate all of the criteria for the trial at once.

**Clinical Accuracy**. To evaluate the *clinical accuracy* of the generated criteria, we run Clinical Trial Parser (FAIR, 2022) on the generated criteria to extract the medical relations and compare them with the relations extracted from the corresponding ground-truth criteria. We evaluate the overlapping of two relation sets by the precision, recall, F1-score, and Jaccard similarity.

**Human Evaluation**. We perform a manual evaluation to compare the generated clinical trial design from our method with the generated by a general LLM. We enlisted the expertise of domain experts to assess and choose the superior output between our method and the LLM's output for a given trial synopsis. This allowed us to collect feedback and calculate the winning rate.

### 4.3 Implementations

To the best of our knowledge, there were no existing algorithms for automatic trial design generation. We thus propose to compare `AutoTrial` with different NLG models: finetuning (FT), prefix-tuning (PT) (Li and Liang, 2021), retrieval-augmented generation (RAG) (Lewis et al., 2020), and contrastive

Table 2: *Automatic evaluation* of eligibility criteria generation results on the test set on the *trial-level*, i.e., compare the concatenated inclusion/exclusion criteria of a trial with the corresponding groundtruth. B1 is short for BLEU-1.

| Method/Scores | Trial level - Inclusion | | | | Trial level - Exclusion | | | |
|---|---|---|---|---|---|---|---|---|
| | B1 | METEOR | ROUGE-L | CIDEr | B1 | METEOR | ROUGE-L | CIDEr |
| GPT2-FT | 9.2 | 23.6 | 7.5 | 0.10 | 7.2 | 8.3 | 2.7 | 0.02 |
| GPT2-RAG | 16.4 | 25.2 | 9.9 | 0.09 | 9.7 | 19.4 | 7.4 | 0.09 |
| GPT2-PT | 20.0 | 19.2 | 22.8 | 0.17 | 16.5 | 15.0 | 12.6 | 0.14 |
| GPT2-SimCTG | 9.8 | 24.8 | 10.6 | 0.11 | 9.4 | 11.4 | 5.5 | 0.06 |
| T5-FT | 21.9 | 29.8 | 18.3 | 0.18 | 13.2 | 13.9 | 8.0 | 0.07 |
| T5-RAG | 22.7 | 28.9 | 16.7 | 0.15 | 14.9 | 11.8 | 7.5 | 0.06 |
| T5-PT | 43.2 | 21.0 | 23.3 | 0.09 | 17.8 | 23.4 | 11.1 | 0.19 |
| T5-SimCTG | 22.1 | 30.3 | 17.6 | 0.17 | 15.8 | 13.1 | 7.7 | 0.06 |
| AutoTrial | **58.7** | **40.8** | **40.6** | **0.24** | **54.4** | **36.3** | **35.3** | **0.33** |

Table 3: *Automatic evaluation* of eligibility criteria generation results on the test set on the *criteria-level*, i.e., compare the generated inclusion/exclusion criteria with the groundtruth one by one. B1 is short for BLEU-1.

| Method/Scores | Criteria level - Inclusion | | | | Criteria level - Exclusion | | | |
|---|---|---|---|---|---|---|---|---|
| | B1 | METEOR | ROUGE-L | CIDEr | B1 | METEOR | ROUGE-L | CIDEr |
| GPT2-FT | 9.0 | 18.1 | 27.0 | 0.54 | 18.2 | 16.4 | 21.4 | 0.47 |
| GPT2-RAG | 19.9 | 17.7 | 28.0 | 0.50 | 13.8 | 17.3 | 22.1 | 0.52 |
| GPT2-PT | 10.1 | 13.9 | 18.3 | 0.23 | 14.1 | 12.3 | 10.9 | 0.13 |
| GPT2-SimCTG | 32.3 | 16.6 | 32.3 | 0.68 | 29.1 | 14.5 | 23.0 | 0.68 |
| T5-FT | 12.8 | 6.8 | 9.4 | 0.11 | 11.2 | 3.9 | 5.1 | 0.04 |
| T5-RAG | 14.7 | 11.3 | 14.2 | 0.30 | 12.1 | 5.3 | 6.8 | 0.09 |
| T5-PT | 22.3 | 9.8 | 15.8 | 0.26 | 10.4 | 15.5 | 9.3 | 0.18 |
| T5-SimCTG | 20.3 | 11.5 | 11.6 | 0.33 | 12.4 | 10.3 | 10.5 | 0.17 |
| AutoTrial | **39.7** | **24.3** | **35.3** | **0.79** | **38.4** | **24.2** | **30.0** | **0.68** |

learning + contrastive search (`SimCTG`) (Su et al., 2022). We choose GPT2 (Radford et al., 2019) and T5 (Raffel et al., 2020) as the backbones. We also compare with GPT-3.5 to verify if general LLMs can generate reasonable eligibility criteria (Ouyang et al., 2022) [1].

For our method, we leverage GPT-2 (Radford et al., 2019) as the backbone model. We set the maximum context length as 1,024. In the pertaining stage, we train the backbone model with a batch size of 64, learning rate 5e-5, weight decay 1e-4, and 5 epochs. In the instruction tuning stage, we train the model with a batch size of 16, learning rate 5e-5, weight decay 1e-5, and 10 epochs.

### 4.4 Exp 1: Generation Quality

**Text quality.** Table 2 shows the automatic evaluation scores with the evaluation done at the trial level. The results show that `AutoTrial` demonstrates superior performance over the baselines in all four metrics (BLEU-1, METEOR, ROUGE-L, and CIDEr) for both inclusion and exclusion criteria. We can draw similar conclusions from Table 3, with the evaluation at the criteria level.

One notable finding is that the performance for inclusion criteria generation is generally better than for exclusion criteria generation. We conjecture that inclusion criteria are presented prior to exclusion criteria in the training data, which may lead to the truncation of the latter due to the model's maximum acceptable length. Besides, errors may accumulate when generating criteria in an autoregressive manner. `AutoTrial` mitigates the order issue credited to the hybrid prompting.

**Clinical accuracy.** We present the clinical accuracy evaluation results in Table 4. As aligned with the automatic evaluation results, `AutoTrial` performs better at criteria generation, with a bigger performance gap. For example, `AutoTrial` is the only method that yields recall above 0.5 (w/ 0.91), F1 above 0.6 (w/ 0.91), and Jaccard scores above 0.4 (w/ 0.84) in inclusion criteria generation. It wins over baselines with a prominent margin in exclusion criteria generation. These results demonstrate that our method can generate criteria accurately aligned with the provided instructions.

We also observe that most methods obtain decent precision and `AutoTrial` has the best performance. It implies a low hallucination rate in our method's generated text because most generated relations

Table 4: *Clinical accuracy* evaluation results of eligibility criteria generation results on the test set. P, R, F1, Jac are short for precision, recall, F1 score, micro-Jaccard score, respectively.

| Type | Method/Score | P | R | F1 | Jac |
|------|--------------|------|------|------|------|
| Inclusion | GPT2-FT | 0.74 | 0.35 | 0.47 | 0.31 |
| | GPT2-RAG | 0.81 | 0.41 | 0.54 | 0.37 |
| | GPT2-PT | 0.75 | 0.45 | 0.56 | 0.39 |
| | GPT2-SimCTG | 0.89 | 0.40 | 0.56 | 0.38 |
| | T5-FT | 0.77 | 0.10 | 0.17 | 0.09 |
| | T5-RAG | 0.82 | 0.13 | 0.22 | 0.12 |
| | T5-PT | 0.74 | 0.17 | 0.27 | 0.16 |
| | T5-SimCTG | 0.68 | 0.04 | 0.08 | 0.04 |
| | AutoTrial | **0.91** | **0.92** | **0.91** | **0.84** |
| Exclusion | GPT2-FT | 0.69 | 0.21 | 0.33 | 0.20 |
| | GPT2-RAG | 0.59 | 0.26 | 0.36 | 0.22 |
| | GPT2-PT | 0.36 | 0.25 | 0.30 | 0.17 |
| | GPT2-SimCTG | 0.80 | 0.23 | 0.36 | 0.22 |
| | T5-FT | 0.24 | 0.03 | 0.06 | 0.03 |
| | T5-RAG | 0.24 | 0.03 | 0.05 | 0.03 |
| | T5-PT | 0.18 | 0.25 | 0.21 | 0.12 |
| | T5-SimCTG | 0.16 | 0.01 | 0.02 | 0.01 |
| | AutoTrial | **0.85** | **0.89** | **0.87** | **0.76** |

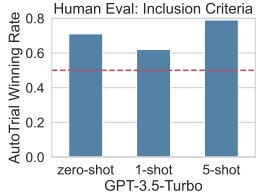
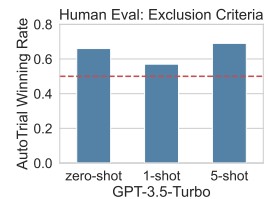

Figure 2: Human evaluations of the winning rate of `AutoTrial` against GPT-3.5 when GPT-3.5 does zero-shot generation (no exemplar), 1-shot, and 5-shot in-context learning.

are also concretely mentioned in the groundtruth eligibility criteria. However, the baselines perform much worse regarding the recall and Jaccard scores. It indicates that `AutoTrial` is advantageous in the high coverage of targeting clinical relations in the generated criteria.

**Human Evaluation.** The human evaluation results are available in Fig. 2, where we identify that our method significantly outperforms the GPT-3.5 baselines, i.e., in over 60% cases, the output criteria from `AutoTrial` are considered better than the from GPT-3.5. This again emphasizes the opportunity of developing expert LLMs that surpass general LLMs at much less cost. It is also interesting that 5-shot GPT-3.5 is worse than the 1-shot and zero-shot ones. We conjecture that GPT-3.5 is impacted by the context that contains irrelevant exemplars when generating for the targeting trials.

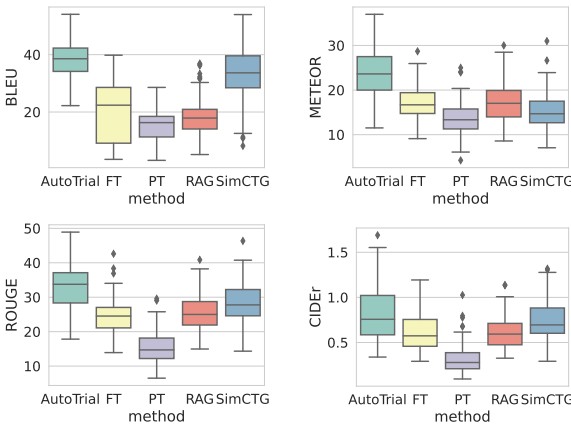

Figure 3: In-depth analysis of generation quality across 100 trial groups divided by the targeting disease. Baselines are based on GPT-2.

## 4.5 Exp 2: In-depth Analysis

**Performance Divergence.** We divided the raw test trials by their target diseases, leading to 100 non-overlapping sets, with each set sharing the same target disease. We then evaluated the generated texts within each subset. We created box plots of the obtained scores (evaluated on the combination of inclusion and exclusion criteria) in Fig. 3.

Our results indicate that `AutoTrial` exhibits superior performance across all metrics. It achieves the highest median performance and has a more stable score distribution, with both a high upper bound and lower bound for all metrics. Among the baseline methods, SimCTG performs the best on three metrics, with the exception of METEOR. However, it should be noted that its worst-case performance was typically much lower than that of most other methods. We also zoom in to show the performances of trials targeting the top eight most frequent diseases/conditions in Fig. 6, where `AutoTrial` consistently wins over all baselines.

**Qualitative Analysis.** We present several qualitative results of our model in Table 5. The model inputs have two parts: manual input and automatically built input, where the manual input is concatenated with the automatic input and passed to the model. Users set up the basic trial information and can opt to offer different instructions for generating criteria. As observed in the first four rows of Table 5, the outputs vary when provided with different instructions for the same trial. Furthermore, it can be observed that the generated outputs are fluent, coherent, and closely resemble the referential manually written criteria.

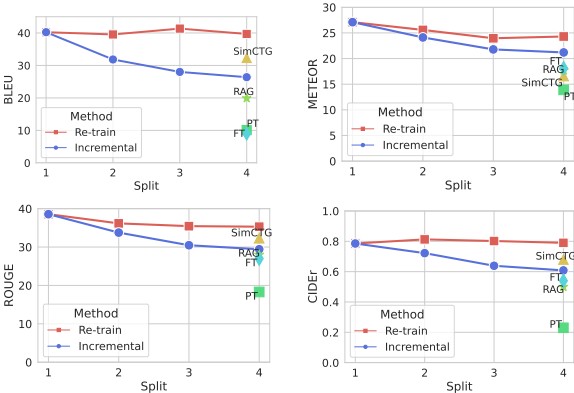

Figure 4: Comparison between two variants of AutoTrial: Re-train and Incremental. The Re-train variant is trained on all subsets, while the Incremental variant updates its knowledge only on new subsets. The scatter plot also includes the performance of four baselines, which are trained on all data.

## 4.6 Exp 3: Incremental Learning

One major merit of AutoTrial is to continually update its internal and external memory without the need of retraining on all collected data. To demonstrate the capability of AutoTrial in continuously updating its knowledge, we designed two variants of our method: Re-train and Incremental. These variants were trained on four subsets of the raw training set: $\{\mathcal{C}_1, \mathcal{C}_2, \mathcal{C}_3, \mathcal{C}_4\}$, with the instruction types being equally assigned and mutually exclusive in each subset. The models encountered the subsets sequentially, with the Re-train model learning by combining all previously seen subsets, e.g., it would be trained and evaluated on $\{\mathcal{C}_1, \mathcal{C}_2\}$ when $\mathcal{C}_2$ is revealed. In essence, Re-train is the theoretic upper bound for all incremental learning methods. In contrast, the Incremental model is also evaluated on $\{\mathcal{C}_1, \mathcal{C}_2\}$ but it would be trained on $\mathcal{C}_2$ only when it is revealed. Additionally, during training, the Incremental model only updates the neural prompting while freezing all other parameters.

We present the results in Fig. 4. The Incremental model demonstrates the capability of mitigating catastrophic forgetting when extended to new data. However, the gap between the two variants expands over time. The Incremental model decays to the level of the best baseline after being updated on the fourth subset when the total number of instructions is $4\times$ more than in the first subset. We hence suggest incrementally updating AutoTrial until the new instructions reach around $3\times$ more than the last fully retrained checkpoint to

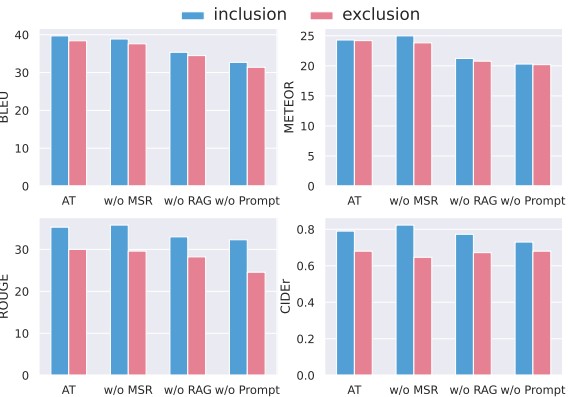

Figure 5: Ablation experiments of AutoTrial when removing one module. AT: the original version; w/o MSR: without the multi-step reasoning supervision; w/o RAG: without the retrieval-augmented generation; w/o Prompt: without the neural prompting.

reach a trade-off between utility and cost.

## 4.7 Exp 4: Ablation Study

We conducted an ablation study (shown in Fig. 5) to compare the original version of AutoTrial with its variants when removing certain components: the multi-step reasoning supervision (w/o MSR), the retrieval-augmented generation (w/o RAG), and the neural prompting (w/o Prompt). The results show that both RAG and Prompt have a significant impact on the final performance. MSR performs similarly on inclusion criteria compared to the original version but has worse results on exclusion criteria. Despite this, MSR is ultimately retained in the final model as it produces more balanced results among inclusion and exclusion criteria and also provides insight into the model's reasoning path, making it more interpretable.

## 5 Conclusion

In summary, this paper presents AutoTrial that uses language models to aid in the design of clinical trial protocols. Our method is able to generate high-quality criteria texts that are fluent, coherent, and clinically accurate, by using a combination of controllable generation, scalable knowledge incorporation, and multi-step reasoning. This can potentially reduce the risk of clinical trial failure by ensuring that trials are properly designed and have sufficient power to evaluate proposed therapies.

## Limitations

The proposed method, AutoTrial, is a valuable tool for designing clinical trials by providing con-

trollable generation under instructions, scalable knowledge incorporation, and multi-step reasoning. However, it is important to note that one limitation of the method is that it is dependent on the quality of data used to train the language model. If the clinical trial database used to train the model contains biases or inaccuracies, these limitations may be present in the generated criteria texts. To ensure the quality of the generated criteria texts, it is crucial to use high-quality, accurate, and up-to-date data to train the language model, which can be achieved by regularly updating the clinical trial databases used for training.

Additionally, the method may not be able to account for unexpected or rare side effects or issues that may occur during the trial, which may impact the safety and efficacy of the proposed treatment. It is important to note that `AutoTrial` should be considered a supportive tool for designing clinical trials and the final decision should always be made by human clinicians. The tool can aid in identifying relevant trials and generating high-quality criteria texts, but ultimately, it is the responsibility of the clinician to evaluate the overall design and safety of the trial, taking into account the unique characteristics and needs of the trial population. The tool should be used as an aid in the design process, but not as a replacement for the expertise and judgment of human clinicians.

## Acknowledgments

This work was supported by NSF award SCH-2205289, SCH-2014438, and IIS-1838042.

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

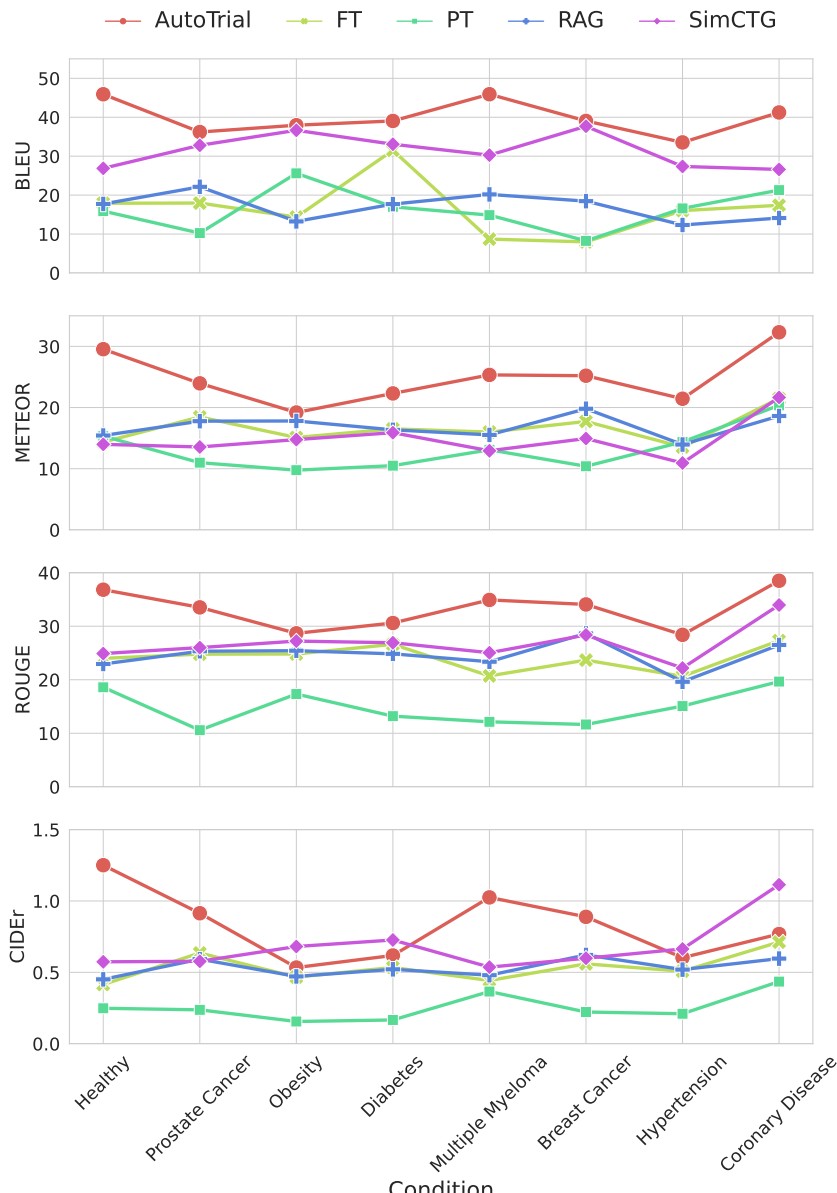

Figure 6: The generation quality across the trials targeting to the top-8 most frequent diseases/conditions.

Table 5: Qualitative generation results of `AutoTrial` for criteria generation under instructions. **Manual Input**: the context textual inputs offered by users, where trial setups $\mathbf{x}_s$ are shared for the same trial and instructions $\mathbf{x}_r$ are specific to each criteria; **Automatically Built Input**: the reference criteria automatically retrieved and built as the input $\mathbf{x}_e$ for `AutoTrial`; **Output**: results generated by `AutoTrial`; **Groundtruth**: the corresponding criteria written by human clinicians in the original trial documents. The **Manual Input** and **Automatically Built Input** will be concatenated as the final input. yellow highlights the *instruction* tokens; green highlights the *setup* tokens; blue highlights the *reference* tokens; red highlights the *target* tokens. Reference texts are truncated in the middle due to the limited space.

| Manual Input | Automatically Built Input | Output | Groundtruth |
|---|---|---|---|
| <instr> <bmi> </instr> <title> A Single-dose and Multiple-dose Study to Evaluate the Pharmacokinetics and Pharmacodynamics of DBPR108 Tablets in Type 2 Diabetes Mellitus Patients <disease> Type 2 Diabetes Mellitus <treatment> DBPR108 tablets | <ref> <inc> subjects with bmi of 20-45 kg/m2 ... <exc> severe gastrointestinal diseases: active ulcer, gastrointestinal or rectal bleeding, active inflammatory bowel syndrome, biliary duct obstruction, active gastritis that is not controlled by medication, etc. </ref> | <incs> <inc> Body mass index 19 to 35 kg/m$^2$, inclusive </incs> | Body mass index (BMI) within the range of 19-35 kg/m$^2$ (inclusive), BMI = weight (kg) / height$^2$ (m$^2$) |
| <instr> <nyha> </instr> <title> A Single-dose and Multiple-dose Study to Evaluate the Pharmacokinetics and Pharmacodynamics of DBPR108 Tablets in Type 2 Diabetes Mellitus Patients <disease> Type 2 Diabetes Mellitus <treatment> DBPR108 tablets | <ref> <inc> subjects with bmi of 20-45 kg/m2 <inc> subjects with 7% $\leq$ hba1c $\leq$ 10% who have been diagnosed with t2dm at least 8 weeks ... <exc> severe gastrointestinal diseases: active ulcer, gastrointestinal or rectal bleeding, active inflammatory bowel syndrome, biliary duct obstruction, active gastritis that is not controlled by medication, etc. </ref> | <excs> <exc> Heart failure (NYHA class III and IV) </excs> | Serious dysrhythmias, obvious left ventricular dysfunction, New York Heart Association (NYHA) functional class III or IV |
| <instr> <sbp> </instr> <title> A Single-dose and Multiple-dose Study to Evaluate the Pharmacokinetics and Pharmacodynamics of DBPR108 Tablets in Type 2 Diabetes Mellitus Patients <disease> Type 2 Diabetes Mellitus <treatment> DBPR108 tablets | <ref> <inc> subjects with bmi of 20-45 kg/m2 ... <exc> severe gastrointestinal diseases: active ulcer, gastrointestinal or rectal bleeding, active inflammatory bowel syndrome, biliary duct obstruction, active gastritis that is not controlled by medication, etc. </ref> | <excs> <exc> Subject has hypotension (systolic blood pressure < 90 mmHg) or hypertension (systolic blood pressure $\geq$ 140 mmHg or diastolic blood pressure $\geq$ 90 mmHg) at screening </excs> | Uncontrolled hypertension, systolic pressure $\geq$160 mmHg or diastolic pressure $\geq$100 mmHg |
| <instr> <cqt> </instr> <title> A Single-dose and Multiple-dose Study to Evaluate the Pharmacokinetics and Pharmacodynamics of DBPR108 Tablets in Type 2 Diabetes Mellitus Patients <disease> Type 2 Diabetes Mellitus <treatment> DBPR108 tablets | <ref> <inc> subjects with bmi of 20-45 kg/m2 ... <exc> severe gastrointestinal diseases: active ulcer, gastrointestinal or rectal bleeding, active inflammatory bowel syndrome, biliary duct obstruction, active gastritis that is not controlled by medication, etc. </ref> | <excs> <exc> Patients who had QTc interval $\geq$ 450 ms in males or $\geq$ 470 ms in females </excs> | Patients who have the second or third degree atrioventricular block, long Q-T syndrome, or QTc>500 ms without cardiac pacemaker |
| <instr> <life_expectancy> </instr> <title> Gefitinib Combined With Chemotherapy or Antiangiogensis in Patients With Bim Deletion or Low EGFR Mutation Abundance <disease> Non-small-cell Lung Cancer <treatment> Gefitinib, pemetrexed or gemcitabine plus carboplatin, bevacizumab | <ref> <inc> female patients with reproductive potential must have a negative serum pregnancy test within 72 hours prior to start of study medication. all female patients of childbearing potential, and all male patients, ... <exc> known brain metastases (in case of clinical signs or symptoms of brain metastases radiological evaluation is mandatory). </ref> | <incs> <inc> Life expectancy $\geq$ 12 weeks </incs> | Life expectancy of at least 12 weeks |
| <instr> <age> </instr> <title> Nutrient Synergy in Beef and Stimulation of Protein Synthesis in Elderly <disease> Healthy <treatment> 3 ounces of cooked, 85% lean ground beef, 20 grams beef protein isolate | <ref> <inc> bmi 18.5 - 29.9 kg/m2 ... <exc> self-reported malabsorption (e.g. difficulty digesting or absorbing nutrients from food, potentially leading to bloating, cramping or gas) </ref> | <incs> <inc> Aged 60 years or older </incs> | Age 60 years or older |