# OpenReview forum: "AutoTrial: Prompting Language Models for Clinical Trial Design"
_EMNLP/2023/Conference — EMNLP 2023 Main_

### Official Review · Reviewer_JAD5 · 2023-08-02

**Soundness:** 3

**Excitement:**

3: Ambivalent: It has merits (e.g., it reports state-of-the-art results, the idea is nice), but there are key weaknesses (e.g., it describes incremental work), and it can significantly benefit from another round of revision. However, I won't object to accepting it if my co-reviewers champion it.

**Paper Topic And Main Contributions:**

This paper studies generative language models for designing clinical trials, focusing on inclusive and exclusive criteria generation. The authors propose a novel prompting-based framework that they train from scratch using existing clinical trial announcements.  The model outputs both clinical trial criteria and the reasoning steps used to generate the criteria.

**Reasons To Accept:**

- important medical application
- good performance
- several analyses are performed to give insight into the results

**Reasons To Reject:**

- The paper is quite dense and hard to follow (for instance, the neural prompt section is quite difficult for readers who aren't prompting experts)
- It's not clear if the data and code will be made available
- There are no details about the implementation and experimental setup: nb of runs, hyperparameters,...

**Reproducibility:**

1: Could not reproduce the results here no matter how hard they tried.

**Reviewer Confidence:**

1: Not my area, or paper was hard for me to understand. My evaluation is just an educated guess.

---

> ### Author Rebuttal · Authors · 2023-08-26
>
> ### Q1 Presentation
>
> > this paper is quite dense and hard to follow
>
> We have made a comprehensive edits of the method section to make sure a better readability. Specifically, we follow the suggestions from Reviewer #AGTQ to add more examples for the introduced components in the prompt.
>
>
>
> ### Q2 Data and code availability
>
> All the data we use are available to download from the CT.GOV database from https://aact.ctti-clinicaltrials.org/data_dictionary. We will release the preprocessed data and the model code in the later stage.
>
>
>
> ### Q3 Implementation details
>
> >  no details about the implementation and experimental setup
>
> We have added the implementation details of our method in Section 4.3 of the new version for better reproducibility.

---

### Official Review · Reviewer_ZRCV · 2023-08-05

**Typos Grammar Style And Presentation Improvements:** sec 4.3 baseline methods
**Soundness:** 4

**Excitement:**

4: Strong: This paper deepens the understanding of some phenomenon or lowers the barriers to an existing research direction.

**Paper Topic And Main Contributions:**

The paper proposes a method named AutoTrial to aid the design of clinical eligibility criteria using language models. The method contains three main technical features: instruction prompt tuning for controllable generation, scalable and efficient knowledge incorporation via in-context learning and multi-step reasoning. Extensive experiments show that this method outperforms many strong baselines by a large margin, and the generated criteria texts are fluent, coherent, clinically accurate and of high-quality.

**Reasons To Accept:**

1. This work shows the retrieved criteria from relevant trails as the exemplars could be used to improve the quality of generation. Ane the in-context exemplar serves as a good template for the model's multi-step reasoning outputs.
2. A discrete and neural prompting approach is proposed to better solve the clinical trial design problem and the method outperforms many strong baselines.
3. Ablation has been shown to justify the importance of RAG and Prompt module, and the MSR is shown to contribute in the model interpretability.

**Reasons To Reject:**

In the clinical trial setting, the coverage of the external knowledge base may not be great enough for all the possible queries. If the retrieval confidence is not good enough, how to facilitate the in-context learning abilities of language models?

**Reproducibility:**

4: Could mostly reproduce the results, but there may be some variation because of sample variance or minor variations in their interpretation of the protocol or method.

**Reviewer Confidence:**

4: Quite sure. I tried to check the important points carefully. It's unlikely, though conceivable, that I missed something that should affect my ratings.

---

> ### Author Rebuttal · Authors · 2023-08-26
>
> ### Q1 Retrieval component performance
>
> > the coverage of the external knowledge base may not be great enough ...
>
> We agree that retrieval performance is a one of the key factors that impact the performance of retrieval-augmented generation models. In this paper, we add the whole clinicaltrial.gov database as the knowledge store to ensure a comprehensive coverage of all types of clinical trials. In practice, it is usually beneficial to add more available knowledge sources, for example, from scientific publications. It is also useful to clean, filter, and summarize the retrieved knowledge to make sure a higher recall of the needed evidence. All these strategies we would like to leave as the future work to explore.

---

### Official Review · Reviewer_AGTQ · 2023-08-13

**Soundness:** 3

**Excitement:**

3: Ambivalent: It has merits (e.g., it reports state-of-the-art results, the idea is nice), but there are key weaknesses (e.g., it describes incremental work), and it can significantly benefit from another round of revision. However, I won't object to accepting it if my co-reviewers champion it.

**Paper Topic And Main Contributions:**

The paper introduces a novel task: the automation of clinical trial design, with a specific focus on eligibility criteria. The objective is to potentially aid the design of clinical trial protocols. The authors systematically address this task by defining the problem, creating a dedicated dataset, introducing a model termed "AutoTrial," and rigorously assessing its performance.

**Questions For The Authors:**

It would be interesting to see further clarification about the choice of evaluation metrics from the authors.

**Reasons To Accept:**

The paper demonstrates novelty through the introduction of a new task: the automation of clinical trial design through the creation of a language model. This has the potential to aid clinical trial design, thus contributing to the advancement of drug development by increasing the likelihood of successful trials. The motivation for clinical trial protocol design, which is a challenging task, is effectively communicated. Overall, the paper is readable and understandable.  In terms of assessment, the authors evaluated their proposed model, considering a comprehensive array of baseline models and employing various metrics to portray its effectiveness. Given the novelty of this task, it is worth considering whether the chosen metrics provide the most optimal evaluation for this specific context, although seemed reasonable.




**Reasons To Reject:**

While the authors have motivated the need for a clinical trial automation process, the authors should be more careful in their word choice. A significant example to this end: it was mentioned in the first paragraph of the introduction, "one area where generative LLMs have shown significant potential is in clinical trial protocol design" without any reference which contradicts the paper's main claim that they are "first to develop LLMs focusing on trial design". The authors also pointed out that the models often lack the capability of "ability to adapt expert instructions" in the introduction without any reference or evidence to it. It is crucial to demonstrate how the generic models fail or at least provide a reference to this claim.

The authors are missing a number of key details in Section 3 , making it difficult to reproduce the paper's results. An example of this is in the pre-training of AutoTrial,  the authors used title, disease, treatment, etc. as the input (i.e. trial setup), however, did not mention the maximum length considered for the input. Generally, clinical trial documents are lengthy documents, and it is of utmost importance in this scenario.  While the importance of the "maximum acceptable length" parameter is acknowledged in Section 4.4, attributing potential performance discrepancies between exclusion and inclusion criteria to truncation, no discussion is presented on the length selected. Additionally, the paper lacks in-depth details about the model used, describing it merely as a "decoder-based causal language modeling architecture."

The problem setup in Section 3.1 is not as clearly demonstrated as it could be. It would be highly beneficial if the different input attributes, such as 'targeting instruction,' 'reasoning steps,' and 'textual description of the given instruction,' were illustrated with a sample input. Given that the paper's primary focus revolves around 'the criteria of trials,' a list of potential criteria (e.g., age, gender, BMI, etc) for the eligibility section (inclusion/exclusion) would greatly aid the reader in comprehending the core concepts.

In summary, while the authors have effectively emphasized the need for clinical trial automation, they should revise the article providing missing details, particularly regarding model parameters, model specifications, and sample input demonstrations, which would greatly enhance the clarity and reproducibility of the paper.





**Reproducibility:**

2: Would be hard pressed to reproduce the results. The contribution depends on data that are simply not available outside the author's institution or consortium; not enough details are provided.

**Reviewer Confidence:**

4: Quite sure. I tried to check the important points carefully. It's unlikely, though conceivable, that I missed something that should affect my ratings.

**Typos Grammar Style And Presentation Improvements:**

There are a few grammatical mistakes and reference issues. For example, in section 3.3 the authors referred to the dataset as section 3.1 which will 4.1. There is a spelling mistake in the title of section 4.3 ( 'mehtods'). There are styling issues throughout the paper while referring to different sections.

---

> ### Author Rebuttal · Authors · 2023-08-26
>
> ### Q1 Clarity
> >  the authors should be more careful in their word choice
>
> We acknowledge that the presentation of this paper can be further improved. For the issue mentioned by the reviewer, we rephrase the parts describing the challenges for AI for trial design with the following three main points:
>
> - **Comprehending instructions**: LLMs need to comprehend the input trial synopsis and adapt to expert inputs to generate precise eligibility criteria.
> - **Referring to prior studies**: LLMs need to be able to refer to prior studies for better generation.
> - **Rationalizing the generation**: LLMs need to offer the rationale behind the generated criteria so as to be trusted by human experts in practice.
>
> ### Q2 Experiment setups & Technical details
>
> > missing a number of key details in Section 3
>
> In Section 3, we mainly describe the pipeline of AutoTrial. We improve the paper presentation to make it more clear for reading and reproducing.
>
> - We add the example of the input components in the problem setup (Section 4.1) for better understanding.
> - We add the implementation details of our method in Experiment section 4.3, including the maximum context length, batch size, etc
>
> We also make a comprehensive polish of the method section for better clarify in the new version.
>
>
>
> ### Q3 Choice of evaluation metrics
>
> > It would be interesting to see further clarification about the choice of evaluation metrics
>
> In Section 4.2, we describe the three evaluation strategies: automatic evaluation, clinical evaluation, and human evaluation. We believe the first and the last are both standard evaluation metrics for general text generation tasks. Automatic evaluation involves a series of text overlapping measurements like CIDEr, ROUGE, etc., that computes how similar the generated criteria are to the target criteria written by human experts. Human evaluation involves the manual check of how well the generated criteria approximate the target criteria as a remedy to the automatic evaluation. We further add a clinical accuracy evaluation to check if the generated criteria describe the same target as the target criteria under the clinical context, e.g., if both criteria describe the same requirement for age and maybe in different ways. It is important because automatic evaluation may overestimate the textual descriptions while neglecting the semantic meaning of the criteria.

---

### Meta-Review · Area_Chair_AL83 · 2023-09-04

**Recommendation:** 4
**Confidence:** 3

**Metareview:**

The reasons to accept, as noted by the reviewers, are mainly in the areas of novelty and originality and include:

- Novelty through the introduction of a new task.
- Motivation for the application is effectively communicated.
- The authors evaluated their proposed model using a comprehensive array of baseline models and various metrics to portray its effectiveness (including human evaluation).
- The method outperforms many strong baselines.
- The ablation study provides important information on the contribution of different components.

The areas for improvement are mainly in terms of clarity and the need to add missing details, especially in Section 3:
- Missing a number of key details in Section 3, which describes the pipeline. In their rebuttal, the authors commit to "add the example of the input components in the problem setup (Section 4.1) for better understanding and add the implementation details of our method in Experiment section 4.3, including the maximum context length, batch size, etc"
-  The coverage of the external knowledge base may not be great enough for all the possible queries.
- Certain sections may be hard to follow (for instance, the neural prompt section is quite difficult for readers who aren't prompting experts)
- It's not clear if the data and code will be made available (the authors note that the data is available now and that the code will be made available in the future).

These can be addressed through revision of the writing and inclusion of further detail, which the authors have committed to provide.

One specific limitation raised by reviewer AGTQ that the authors do not address in their response is as follows: 'it was mentioned in the first paragraph of the introduction, "one area where generative LLMs have shown significant potential is in clinical trial protocol design" without any reference which contradicts the paper's main claim that they are "first to develop LLMs focusing on trial design". ' In my view, it is crucial to appropriately address this point in order to convey the novelty of the contribution presented in the current paper.

The authors should pay careful attention to implementing the suggested improvements to ensure a clearer and more accessible presentation of their work.

---

### Decision · Program_Chairs · 2023-10-07

**Decision:**

Accept-Main

**Comment:**

The reasons to accept, as noted by the reviewers, are mainly in the areas of novelty and originality and include:

- Novelty through the introduction of a new task.
- Motivation for the application is effectively communicated.
- The authors evaluated their proposed model using a comprehensive array of baseline models and various metrics to portray its effectiveness (including human evaluation).
- The method outperforms many strong baselines.
- The ablation study provides important information on the contribution of different components.

The areas for improvement are mainly in terms of clarity and the need to add missing details, especially in Section 3:
- Missing a number of key details in Section 3, which describes the pipeline. In their rebuttal, the authors commit to "add the example of the input components in the problem setup (Section 4.1) for better understanding and add the implementation details of our method in Experiment section 4.3, including the maximum context length, batch size, etc"
-  The coverage of the external knowledge base may not be great enough for all the possible queries.
- Certain sections may be hard to follow (for instance, the neural prompt section is quite difficult for readers who aren't prompting experts)
- It's not clear if the data and code will be made available (the authors note that the data is available now and that the code will be made available in the future).

These can be addressed through revision of the writing and inclusion of further detail, which the authors have committed to provide.

One specific limitation raised by reviewer AGTQ that the authors do not address in their response is as follows: 'it was mentioned in the first paragraph of the introduction, "one area where generative LLMs have shown significant potential is in clinical trial protocol design" without any reference which contradicts the paper's main claim that they are "first to develop LLMs focusing on trial design". ' In my view, it is crucial to appropriately address this point in order to convey the novelty of the contribution presented in the current paper.

The authors should pay careful attention to implementing the suggested improvements to ensure a clearer and more accessible presentation of their work.